# PastNet: Introducing Physical Inductive Biases for Spatio-temporal Video Prediction

Hao Wu*
School of Computer Science and
Technology, University of Science and
Technology of China
Hefei, China
wuhao2022@mail.ustc.edu.cn

Fan Xu*
School of Computer Science and
Technology, University of Science and
Technology of China
Hefei, China
markxu@mail.ustc.edu.cn

Chong Chen
Terminus Group
Beijing, China
chenchong.cz@gmail.com

Xian-Sheng Hua
Terminus Group
Beijing, China
huaxiansheng@gmail.com

Xiao Luo†
Department of Computer Science,
University of California
Los Angeles, USA
xiaoluo@cs.ucla.edu

Haixin Wang†
Department of Computer Science,
University of California
Los Angeles, USA
whx@cs.ucla.edu

## ABSTRACT

In this paper, we investigate the challenge of spatio-temporal video prediction, which involves generating future videos based on historical data streams. Existing approaches typically utilize external information such as semantic maps to enhance video prediction, which often neglect the inherent physical knowledge embedded within videos. Furthermore, their high computational demands could impede their applications for high-resolution videos. To address these constraints, we introduce a novel approach called Physics-assisted Spatio-temporal Network (PastNet) for generating high-quality video prediction. The core of our PastNet lies in incorporating a spectral convolution operator in the Fourier domain, which efficiently introduces inductive biases from the underlying physical laws. Additionally, we employ a memory bank with the estimated intrinsic dimensionality to discretize local features during the processing of complex spatio-temporal signals, thereby reducing computational costs and facilitating efficient high-resolution video prediction. Extensive experiments on various widely-used datasets demonstrate the effectiveness and efficiency of the proposed PastNet compared with a range of state-of-the-art methods, particularly in high-resolution scenarios. Our code is available at https://github.com/easylearningscores/PastNet.

## CCS CONCEPTS

• **Computing methodologies** → *Scene understanding*.

## KEYWORDS

Spatiotemporal predictive learning, Physical Inductive Biases, high-resolution video prediction

## 1 INTRODUCTION

Spatial-temporal forecasting has emerged as a significant area of interest within the multimedia research community [8, 26, 53, 58]. Among numerous practical problems, the objective of video prediction is to generate future video frames by leveraging historical

*Equal contribution.
†Corresponding authors.

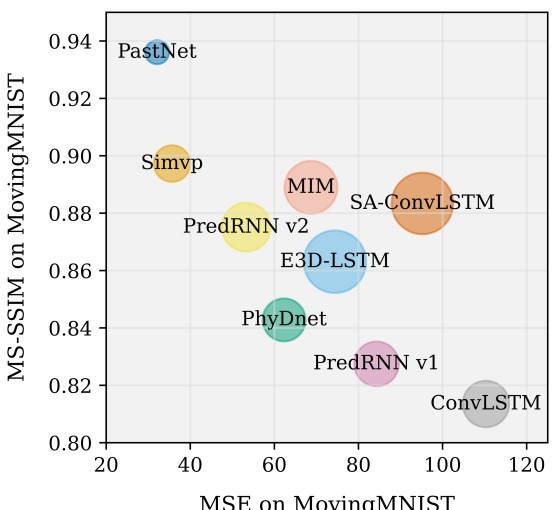

**Figure 1: Performance comparison of video prediction methods on MovingMNIST. PastNet outperforms previous models in training time and image quality, achieving the lowest MSE, highest SSIM, and shortest training time over 100 epochs. The small bubble for PastNet indicates minimal training time.**

frames [7, 49]. This problem bears considerable relevance to an array of applications, including human motion prediction [1], climate change analysis [39], and traffic flow forecasting [48].

In literature, a multitude of methods have been devised for efficacious video prediction, integrating deep neural networks to capture complex correlations within spatio-temporal signals [5]. Early approaches [29, 39, 44, 51] amalgamate convolutional neural networks (CNNs) and recurrent neural networks (RNNs) to extract features from RGB frames and predict future trends, respectively. Several techniques also employ deep stochastic models to generate video prediction while taking into account diverse potential results [1, 13, 43, 52, 53]. More recently, a range of algorithms has sought to enhance video prediction by incorporating external information such as optical flow, semantic maps, and human posture data [22, 27, 32, 45]. For example, SADM [2] combines semantic

maps with flow fields to supply contextual information exhibiting superior compatibility. Nevertheless, these external inputs could not be readily available in practical situations [20, 32, 50]. In light of this consideration, a recent study [14] demonstrates that a basic CNN-based model can achieve state-of-the-art performance through end-to-end optimization.

Despite their remarkable achievements, the performance of existing approaches remains far from satisfactory for the following reasons: (1) **Neglect of Underlying Physical Principles.** Current methods typically employ deep neural networks to extract information from spatial space and the associated visual domains [14, 22, 27, 45]. However, video frames could be governed by underlying physical principles, such as partial differential equations (PDEs) [16, 28, 34]. For instance, climate videos are typically dominated by high-order equations. As a consequence, it is anticipated to explore these physical principles for effective video prediction. (2) **Low Efficiency.** As a dense prediction problem [36], the scalability of neural network models is critical for high-resolution video prediction [5]. Regrettably, the majority of existing models rely on complex neural networks, such as deep CNNs and Vision Transformers [37, 56], which entail significant computational costs and render them unsuitable for large-scale high-resolution videos.

To address these concerns, this paper introduces a novel approach called Physics-assisted Spatio-temporal Network (PastNet) for high-quality video prediction. The croe of our PastNet is to introduce inductive physical bias using the data itself, which holds the potential to solve underlying PDEs. Specifically, we introduce a convolution operator in the spectral space that initially transfers video frames into the Fourier domain, followed by efficient parallelizable channel fusion [17]. Subsequently, we employ an inverse Fourier transform to generate the outputs. Furthermore, to enhance efficiency for high-resolution video implementation, our PastNet introduces a discrete spatio-temporal module, which not only estimates intrinsic dimensionality but also introduces memory banks to discretize local features during the processing of complex spatio-temporal signals, replacing local features with their nearest queries from the memory bank. Finally, a deconvolution decoder is incorporated to output the predictions, which are combined with outputs from the spectral space. Comprehensive experiments on various benchmark datasets substantiate the effectiveness and efficiency of our proposed PastNet. A glimpse of the compared results by various approaches is provided in Figure 1 and we can observe the huge superiority of our PastNet on MovingMNIST. Our main contributions can be summarized as follows:

- New Perspective. We open up a new perspective to connect spatio-temporal video prediction with physical inductive biases, thereby enhancing the model with data itself.
- Novel Methodology. Our PastNet not only employs a convolution operator in the spectral space to incorporate physical prior but also discretizes local features using a memory bank with the estimated intrinsic dimensionality to boost efficiency for high-resolution video prediction.
- High Performance and Efficiency. Comprehensive experiments on a variety of datasets demonstrate that the PastNet exhibits competitive performance in terms of both effectiveness and efficiency.

## 2 RELATED WORK

### 2.1 Spatio-temporal Video Prediction

Video prediction has emerged as an essential topic within the multimedia research community, and numerous methods have been proposed to address this challenge. Initial studies frequently examine spatio-temporal signals extracted from RGB frames [29, 39, 44, 47, 51]. For instance, ConvLSTM [39] employs convolutional neural networks (CNNs) to encode spatial data, which is subsequently integrated with an LSTM model to capture temporal dependencies. PredNet [29] draws inspiration from neuroscience, enabling each layer to make local predictions for video sequences. MCnet [44] introduces multiple pathways to encode motion and content independently, which are combined into an end-to-end framework. Various approaches strive to merge video prediction with external information from optical flow, semantic maps and human posture data [22, 27, 32, 45]. As a representative unsupervised method, DVF [27] predicts missing frames using masked ones. HVP [22] treats this problem as video-to-video translation from semantic structures, inspired by hierarchical models. SADM [2] combines semantic maps and flow fields to provide more compatible contextual information. However, this external information could be inaccessible in real-world applications [20, 32, 50]. Furthermore, the efficiency and effectiveness of current solutions remain suboptimal for high-resolution videos. To surmount these obstacles, we propose a novel method that incorporates both physical inductive biases and quantization operations for high-quality video prediction.

### 2.2 Physics-Informed Machine Learning

Various machine learning problems can benefit from the incorporation of physical knowledge [21]. Modern physics-informed machine learning approaches can leverage knowledge from three aspects, i.e., observational biases, inductive biases, and learning biases. Observational biases primarily arise from the data itself [24, 30, 55], offering a range of data augmentation strategies to expand datasets. Inductive biases guide the specific design of neural networks such as graph neural networks [3] and equivariant networks [9], which possess properties of respecting additional symmetry groups [3]. Learning biases pertain to the process of imposing distinct constraints by incorporating loss objectives during optimization [15], adhering to the principles of multi-task learning. For instance, physics-inspired neural networks [35] (PINNs) typically include constraints related to derivatives from PDEs, resulting in superior performance in modeling dynamical systems and predicting molecular properties. To bolster predictive performance, our PastNet introduces inductive biases from the underlying PDEs of video frames by integrating learnable neural networks in the Fourier domain.

### 2.3 Data Compression

Compressing large-scale data is essential for enhancing the efficiency of both training and inference processes [31]. Learning to hash is a widely employed technique that accomplishes this by mapping continuous vectors to compact binary codes while preserving similarity relationships, which achieves extensive progress in approximate nearest neighbor search [4]. Another line to address this challenge is neural quantization. For instance, multi-codebook

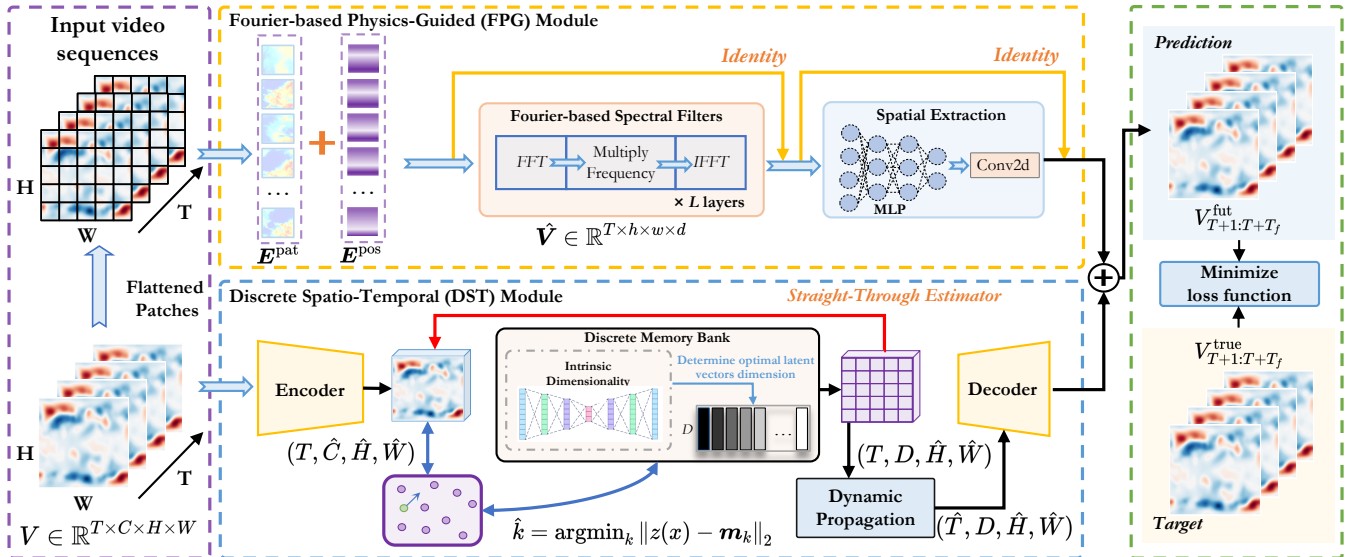

**Figure 2: An overview of the proposed PastNet, which consists of a Fourier-based Physics-guided (FPG) and a Discrete Spatio-temporal (DST) module. The PFI module first divides video frames into non-overlapping patches and introduce Fourier-based spectral filter with the introduction of physical biases. Then, its also extract spatial signals with convolutional neural networks. The DST module is an encoder-decoder architecture, which introduces a memory bank to discretize local features.**

quantization [59] is akin to the process of k-means clustering, which stores centroids and assignments in the codebooks. Recently, VQ-VAE [41] has integrated neural quantization with auto-encoders, using discrete codes to reconstruct input images and leading to efficient models for large-scale applications. VQ-VAE has been successfully applied in various scenarios, including video generation [54], image inpainting [33], and semantic communication [19]. In this paper, our DST module is inspired by VQ-VAE, discretizing local features to improve efficiency in high-resolution video prediction.

represents a dynamic physical system in the temporal domain, consisting of $T$ time steps, denoted as $V_{1:T} = \{V_1, \cdots, V_T\}$. Each snapshot captures $C$ color space measurements over time at all locations within a spatial region, represented by an $H \times W$ grid. From a spatial viewpoint, the observation of these $C$ measurements at any specific time step $i$ can be depicted as a tensor, $V_i \in \mathbb{R}^{C \times H \times W}$. Our objective is to leverage spatio-temporal data to deduce underlying physical priors and integrate feature representation learning in both spatial and temporal dimensions and predict the most probable future sequence of length $T_f$, denoted as $V^{fut}_{T+1:T+T_f} = \{V_{T+1}, \cdots, V_{T+T_f}\}$.

## 3 METHODOLOGY

### 3.1 Overview

This paper studies the problem of spatio-temporal video prediction and existing solutions usually neglect underlying physical principles and suffer from larger computational space. To tackle this, our proposed PastNet introduces both *Fourier-based physics-guided (FPG) module* and *discrete spatio-temporal (DST) module* for effectively and efficiently video prediction as in Figure 2. In particular, our FPG module first divides frame into non-overlapping patches and then introduce a Fourier-based *priori* spectral filters with the introduction of physical inductive biases. Then, our DST module not only estimates intrinsic dimensionality but also introduces a discrete memory bank to effectively and efficiently capture spatio-temporal signals. Following is the problem definition and the detailed description of these key components of our proposed PastNet.

**Problem Definition.** To enhance clarity, we offer a comprehensive explanation of the relevant concepts. Assume a video trajectory

### 3.2 Fourier-based Physics-Guided (FPG) Module

Our primary insight focuses on utilizing prior physical signals to achieve effective spatio-temporal video prediction. In fact, spatio-temporal data is often subject to complex, high-dimensional non-linear physical equations (e.g., Navier-Stokes equation), which are challenging to capture. Inspired by previous studies [17, 24], we employ spectral methods and develop an algorithm that seamlessly combines trainable neural networks with Fourier-based a *priori* spectral filters. It has been shown that the features transformed through Fourier transformation in the frequency domain correspond precisely with the coefficients of the underlying physical partial differential equation [18, 24]. As a result, we can utilize neural networks to approximate the analytical solution of the latent PDE. To be specific, we initially divide video frames into non-overlapping patches with initialized embeddings and subsequently transform them into a spectral space. The features within the frequency domain are fused, followed by an inverse transformation returning

them to the spatial domain. This innovation supports a physical inductive bias derived from data, demonstrating significant potential for solving PDEs. Then, we introduce our FPG module in detail.

**Embedding Initialization.** Given the input video $V \in \mathbb{R}^{T \times C \times H \times W}$, we extract the high-level learnable representations following ViT [11]. In particular, we divide the frame into non-overlapping $N = HW/hw$ patches of size $h \times w$ and then project them into patch embeddings $E^{pat} \in \mathbb{R}^{T \times h \times w \times d}$, where $d$ denotes the embedding dimension. Position embeddings $E^{pos} \in \mathbb{R}^{h \times w \times d}$ are also applied to get the initial token representation matrix $\hat{V} \in \mathbb{R}^{T \times h \times w \times d}$. In formulation,

$$\hat{V}_t = E_t^{pat} + E^{pos}, \tag{1}$$

where $\hat{V}_t = \hat{V}[t,:,:] \in \mathbb{R}^{h \times w \times d}$ makes up the matrix $\hat{V}$.

**Fourier-based Spectral Filter.** We apply $\hat{V}$ as input to $L$ layers of filters, each layer containing three essential components, i.e., *Fourier transform*, *separate mixing* and *inverse Fourier transform*. Firstly, a 2D fast Fourier transform (FFT) is leveraged to generate the frequency domain token $\mathcal{K}_t \in \mathbb{R}^{h \times w \times d}$ at time step $t$:

$$\mathcal{K}_t(u,v) = \sum_{x=0}^{h-1} \sum_{y=0}^{w-1} \hat{V}_t(x,y) e^{-2\pi i (\frac{u}{h}x + \frac{v}{w}y)}, \tag{2}$$

where $i$ is the imaginary unit, and $u$ and $v$ are the indices of rows and columns in the frequency domain, respectively.

Secondly, the complex-valued token $\mathcal{K} = \mathcal{K}_{1:T}$ is then split into its real and imaginary parts and concatenated along the channel dimension. To enhance the integration of feature information, we utilize token mixing across different channels, which allows for richer Fourier mode representations to emerge through greater fusion of channel-wise signals. It is implemented with separate MLPs for the real and imaginary parts separately as follows:

$$\Re\hat{\mathcal{K}}_t(u,v) = \mathrm{MLP}_{\theta_1}(\Re\mathcal{K}_t(u,v)), \Im\hat{\mathcal{K}}_t(u,v) = \mathrm{MLP}_{\theta_2}(\Im\mathcal{K}_t(u,v)), \tag{3}$$

where $\Re$ and $\Im$ denotes the operator to obtain the real part and imaginary part, respectively. Performing token mixing mixes different modes across the Fourier domain. As the Fourier domain possesses global attributes, it further explore long-range relationships for underlying physical features.

Lastly, the mixed tokens are then transformed back to the spatial domain using the 2D inverse Fourier transform to obtain the output of spectral filter layers $Y \in \mathbb{R}^{T \times h \times w \times d}$ as follows:

$$
\begin{aligned}
Y_t(x,y) = &\frac{1}{hw} \sum_{u=0}^{h-1} \sum_{v=0}^{\frac{w}{2}} \hat{\mathcal{K}}_t(u,v) e^{2\pi i (\frac{u}{h}x + \frac{v}{w}y)} \\
&+ \frac{1}{hw} \sum_{u=0}^{h-1} \sum_{v=\frac{w}{2}+1}^{w-1} \hat{\mathcal{K}}_t(u,v) e^{2\pi i (\frac{u}{h}x + \frac{v-w}{w}y)},
\end{aligned} \tag{4}
$$

where $Y_t = Y[t,:,:] \in \mathbb{R}^{N \times M}$ makes up the matrix $Y$.

**Spatial Extraction.** To better extract the latent spatial information, we introduce classic convolutional neural networks as a supplement of the *priori* spectral filters, which can be formulated as follows:

$$\hat{Y}_t^{FPG} = \mathrm{Tanh}(\mathrm{Conv2d}(\mathrm{MLP}(Y_t) + Y_t)), \tag{5}$$

The convolutional layer is known for its ability to extract features in the spatial domain through its local receptive fields. By leveraging

these learned features as filters in the frequency domain, the convolutional layer can effectively mine potential physical information from the input data. The extracted physical information can significantly enhance the performance of spatiotemporal prediction.

Overall, the Fourier-based physics-guided module transforms the spatial domain of latent physical information into the frequency domain using Fourier transforms, and learns the analytical solutions of PDEs using neural networks. This approach allows PastNet to handle complex geometries and high-dimensional problems.

## 3.3 Discrete Spatio-temporal (DST) Module

Our discrete spatio-temporal (DST) module aims to explore spatio-temporal signals in video frames in an efficient manner. To achieve this, we not only estimate the intrinsic dimensionality for the hidden space, but also introduces a memory bank to vector quantization. In particular, it consists of four different modules, i.e., *encoder*, *intrinsic dimensionality estimation*, *discrete quantization*, *dynamic propagation* and *decoder*. Then, we introduce them in detail.

**Encoder.** The encoder contains $K_e$ ConvNormReLU blocks to capture spatial signals. Given $Z^{(0)} = V$ and an activation function $\sigma$, we have:

$$Z^{(i)} = \sigma\left(\mathrm{LayerNorm}\left(\mathrm{Conv2d}\left(Z^{(i-1)}\right)\right)\right), 1 \le i \le K_e, \tag{6}$$

where $Z^{(i-1)}$ and $Z^{(i)}$ denote the input and output of the $i$-th block with the shapes $(T, C, H, W)$ and $(T, \hat{C}, \hat{H}, \hat{W})$, respectively.

**Intrinsic Dimensionality Estimation.** How to decide the dimensionality of the hidden space remains a challenging problem [6]. In particular, too large dimensionality would bring in redundant computational time and potential overfitting while too small one would underfit data. Here, we turn to Levina–Bickel algorithm [23] to acquire intrinsic dimensionality.

In particular, we start with a large dimensionality followed by mapping $Z^{(K_e)}$ back to the input using a decoder and minimize the reconstruction loss objective as $L_{rec} = ||V - \hat{V}||$ where $\hat{V}$ is the reconstructed frame. Then, we identify the $R$ nearest neighbours for each vector $\mathbf{h}_j \in Z^{(K_e)}$, i.e., $\{\mathbf{h}_{j_1}, \cdots, \mathbf{h}_{j_R}\}$ and calculate the local estimator for the vector as:

$$D_j = \frac{1}{R-2} \sum_{m=1}^{R-1} \log \frac{d(\mathbf{h}_j, \mathbf{h}_{j_R})}{d(\mathbf{h}_j, \mathbf{h}_{j_m})}, \tag{7}$$

where $d(\cdot, \cdot)$ denotes the cosine distance between two vectors. Finally, we take the average all local estimators to generate the final estimator:

$$D = \frac{1}{J} \sum_{j=1}^{J} D_j, \tag{8}$$

where $J$ is the number of vectors in $Z^{(K_e)}$ and $\cdot$ denotes the ceiling function. After generating final estimated optimal dimension, we utilize $D$ as the hidden embedding instead.

**Discrete Quantization.** Previous methods usually process video features directly using spatio-temporal convolution modules. However, directly feeding video features into these modules would bring in huge computational cost. Therefore, we introduce a discrete memory bank to discretize feature vectors, which are constructed by an variational autoencoder [41, 54]. In this way, computational costs can be largely reduced to fit for large-scale video prediction.

In detail, we initialize the memory bank with variational autoencoder. Here, each embedding vector $z$ from $Z^{(K_e)}$ from the output of the encoder is mapped to the nearest point in the memory bank. The number of embeddings in the memory is set to $D^2$ empirically. Given the memory bank with $D^2$ embedding vectors, $\{m_1, \cdots, m_{D^2}\}$, we construct a mapping $VQ$:

$$VQ(z) = m_{\hat{k}}, \quad \text{where} \quad \hat{k} = \text{argmin}_k \|z(x) - m_k\|_2, \quad (9)$$

where each embedding $z$ is concatenated to generate matrix $\bar{Z} = VQ(Z^{(K_e)})$. The mapping connects continuous vectors with given vectors in the memory bank to save the computational cost. Then, to minimize the information loss, we map the concatenated matrix back to the input using a new decoder, i.e., $\tilde{V} = Dec(\bar{Z})$. The whole framework is optimized using the following objective as:

$$\mathcal{L} = \|V - \tilde{V}\| + \left\|\text{sg}\left[\bar{Z}\right] - Z^{(K_e)}\right\|_2^2 + \beta \left\|\bar{Z} - \text{sg}[Z^{(K_e)}]\right\|_2^2, \quad (10)$$

where $\beta$ denotes a parameter to balance these objective and $\text{sg}(\cdot)$ is the stopgradient operator to cut off the gradient computation during back propagation. Here, the first term denotes the reconstruction loss and the last two terms minimize the quantization loss between continuous embedding vectors and their neighbours in the memory bank.

**Dynamic Propagation.** After training the variational autoencoder, we remove the decoder, and then feed the quantized vector into temporal convolution on $T \times D$ channels. In particular, each temporal convolution block involves a bottleneck followed by group convolution operator:

$$Z^{(i)} = \text{GroupConv2d}(\text{Bottleneck}(Z^{(i-1)})), K_e < i \le K_e + K_t, \quad (11)$$

where Bottleneck denotes the 2D convolutional layer with $1 \times 1$ kernel and $K_t$ is the number of blocks. The shape of input $Z^{(i-1)}$ and output $Z^{(i)}$ are $(T, D, \hat{H}, \hat{W})$ and $(\hat{T}, D, \hat{H}, \hat{W})$, respectively.

**Decoder.** Finally, our decoder contains $K_d$ unConvNormReLU blocks to output the final predictions $\hat{Y}^{DST} = Z^{(K_e+K_t+K_d)}$. In formulation, we have:

$$Z^{(i)} = \sigma\left(\text{LayerNorm}\left(\text{unConv2d}\left(Z^{(i-1)}\right)\right)\right),$$
$$K_e + K_t + 1 \le i \le K_e + K_t + K_d, \quad (12)$$

where unConv2d is implemented using ConvTranspose2d [12]. The shape of input $Z^{(i-1)}$ and output $Z^{(i)}$ are $(\hat{T}, D, \hat{H}, \hat{W})$ and $(T, C, H, W)$, respectively.

### 3.4 Framework Summarization

Finally, we combine the output of both FPG and DST modules, which result in the final prediction:

$$\hat{Y}^{final} = \hat{Y}^{FPG} \oplus \hat{Y}^{DST}, \quad (13)$$

where $\oplus$ represents element-wise addition. The whole framework would be optimizing by minimizing the vanilla MSE loss between the predictions and the target frame.

## 4 EXPERIMENT

### 4.1 Experimental Setups

**Datasets.** In this paper, the datasets studied can be classified into two categories from the perspective of PDE modeling or physics

**Table 1: Statistics of the datasets used in the experiments. The number of training and test sets of the dataset are $N\_train$ and $N\_test$ respectively, where the size of each image frame is $(C, H, W)$, and the length of the input and prediction sequences are $T$, $K$ respectively.**

| Dataset | $N\_train$ | $N\_test$ | $(C, H, W)$ | $T$ | $K$ |
|---|---|---|---|---|---|
| MovingMNIST | 9000 | 1000 | (1, 64, 64) | 10 | 10 |
| TrafficBJ | 19627 | 1334 | (2, 32, 32) | 4 | 4 |
| KTH | 108717 | 4086 | (1, 128, 128) | 10 | 20 |
| SEVIR | 4158 | 500 | (1, 384, 384) | 10 | 10 |
| RDS | 2000 | 500 | (3, 128, 128) | 2 | 2 |
| EDPS | 2000 | 500 | (3, 128, 128) | 2 | 2 |
| FS | 2000 | 500 | (3, 128, 128) | 2 | 2 |

equation description: **Non-natural Phenomenon** datasets and **Natural Phenomenon** datasets. The former includes **MovingMNIST** [40], **TrafficBJ** [57], and **KTH** datasets [38]. Although they do not correspond to natural phenomena, the dynamic evolutionary processes expressed in these datasets can still be described by PDEs. The latter includes the **Storm EVent ImagRy (SEVIR)** [42], **Reaction Diffusion System (RDS)**, **Elastic Double Pendulum System (EDPS)**, and **Fire System (FS)** datasets [6], which correspond to natural phenomena such as meteorology, chemical reactions, mechanical vibrations, and fire. These datasets are often used in the study of PDE modeling and the description of physics equations to better understand and predict the evolution of natural phenomena. We conduct experiments on seven datasets for evaluation. Here we summarize the details of the datasets used in this paper, The statistics are shown in the Table 1.

**Evaluation metrics.** We adopt Mean Squared Error (MSE), Mean Absolute Error (MAE), Multi-Scale Structural Similarity (MS-SSIM), Peak Signal-to-Noise Ratio (PSNR), and Learned Perceptual Image Patch Similarity (LPIPS) to evaluate the quality of the predictions. Lower values of MSE, MAE, and LPIPS and higher values of SSIM and PSNR imply better performance.

**Implementation details.** PastNet model features a consistent backbone architecture for all datasets, in which the FPG component is composed of 8-*layer Fourier-based Spectral Filter*. The DSM encoder incorporates 3-*layer convolution block layers* and 3-*layer residual blocks*, while the decoder utilizes 1-*layer convolution block*, 4-*layer residual blocks*, and 2-*layer deconvolution blocks*. All experiments in this paper were conducted on an NVIDIA A100-40GB.

### 4.2 Performance Comparison

We conduct a thorough evaluation of PastNet by comparing it with several baseline models on both non-natural and natural phenomena datasets. This includes competitive RNN architectures such as ConvLSTM [39], PredRNN-V1-2 [47], E3D LSTM [46], SA-ConvLSTM [25], PhyDnet [16], and MIM [48]. We also evaluate state-of-the-art CNN architecture SimVP [14] for non-natural phenomena datasets. For natural phenomenon datasets, we evaluate models that incorporate physical information, such as DLP [10], which uses an advection-diffusion flow model and achieves state-of-the-art performance on the SST dataset, it is commonly used

**Table 2: Quantitative prediction results of PastNet compared to Baselines on various Non-natural Phenomenon datasets. The evaluation metrics selected for this study are MSE ↓, MAE ↓, MS-SSIM ↑, and PSNR ↑, with a lower value (↓) indicating better performance for MSE and MAE, and a higher value (↑) indicating better performance for MS-SSIM and PSNR. The best result is indicated in boldface, while the second-best result is indicated with an underline in the table caption.**

| Method | MovingMNIST | | | | TrafficBJ | | | | KTH | | | |
|---|---|---|---|---|---|---|---|---|---|---|---|---|
| | MSE | MAE | MS-SSIM | PSNR | MSE x 100 | MAE | MS-SSIM | PSNR | MSE | MAE / 10 | MS-SSIM | PSNR |
| ConvLSTM | 105.41 | 188.96 | 0.7498 | 26.27 | 48.45 | 18.921 | 0.9782 | 37.72 | 126.15 | 128.32 | 0.7123 | 23.58 |
| PredRNN-V1 | 81.87 | 147.31 | 0.8697 | 29.99 | 46.49 | 17.784 | 0.9789 | 38.11 | 101.52 | 99.21 | 0.8292 | 25.55 |
| E3D LSTM | 67.25 | 136.99 | 0.8907 | 31.02 | 44.89 | 17.219 | 0.9731 | 38.71 | 86.17 | 85.55 | 0.8663 | 27.92 |
| SA-ConvLSTM | 79.76 | 142.21 | 0.8692 | 30.21 | 43.99 | 17.093 | 0.9672 | 38.98 | 89.35 | 87.20 | 0.8372 | 27.25 |
| PhyDnet | 58.22 | 145.76 | 0.9012 | 32.13 | **42.21** | 16.975 | 0.9821 | **39.94** | 66.95 | 56.73 | 0.8831 | 28.02 |
| MIM | 66.28 | 119.87 | 0.9017 | 31.12 | 43.98 | 16.645 | 0.9712 | 38.99 | 56.59 | 54.86 | 0.8666 | 28.97 |
| PredRNN-V2 | 48.42 | 126.18 | 0.8912 | 33.19 | 43.89 | 16.982 | 0.9723 | 39.02 | 51.15 | 50.64 | 0.8919 | 29.92 |
| SimVP | 32.22 | 90.12 | 0.9371 | 37.17 | 43.32 | 16.897 | 0.9822 | 39.29 | 40.99 | 43.39 | 0.9061 | 33.72 |
| PastNet | **31.77** | **89.33** | **0.9447** | **38.38** | 42.93 | **16.405** | **0.9876** | 39.42 | **33.83** | **35.26** | **0.9279** | **35.28** |

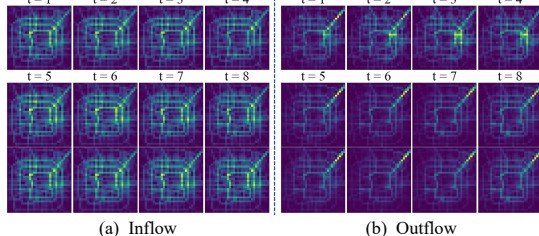

(a) Inflow                    (b) Outflow

**Figure 3: Prediction results on the TrafficBJ dataset. Top: input Traffic flow; Middle: future real Traffic flow; Bottom: PastNet predicted Traffic flow.**

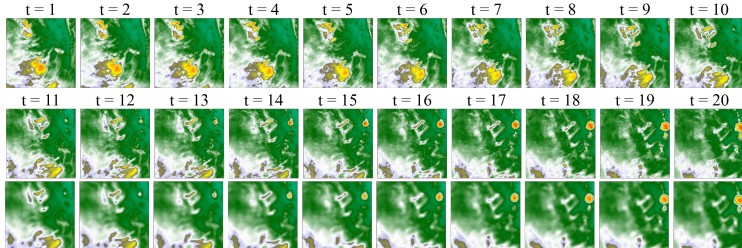

**Figure 4: Example of prediction results on the SEVIR dataset. Top: input weather sequence; Middle: future real weather sequence; Bottom: PastNet predicted weather sequence.**

for generic physical processes. We also evaluate NLDM [6], which combines manifold learning theory and autoencoder to discover fundamental variables hidden in physical experimental data for spatiotemporal prediction, as well as PhyDnet and SimVP. Our evaluation is meticulous to ensure the validity of the results.

Table 2 demonstrates that PastNet outperforms other models on non-natural phenomena datasets. Specifically, PastNet achieves the best MSE and MAE metrics on MovingMNIST, with values that are 20% and 10% lower than SimVP, respectively. Moreover, PastNet also achieves higher MS-SSIM and PSNR metrics than other models, indicating better prediction ability for dynamic changes in videos. On TrafficBJ, while PhyDnet has the best MSE and PSNR performance, PastNet comes in second place and achieves top spot for MAE and MS-SSIM. On KTH, PastNet achieves the best MSE and MAE performance, with values that are 33.6% and 35.8% lower than the second-place MIM model, respectively. Overall, PastNet shows significant advantages over other baseline methods in all evaluation metrics.

Results in Table 3 show that PastNet achieves the best performance in most evaluation metrics for all natural phenomena datasets. On the SEVIR dataset, PastNet achieves the lowest MSE and MAE scores, significantly better than other methods such as DLP, PhyDnet, and NLDM. On the RDS dataset, PastNet achieves the lowest MSE x 100 score, significantly better than other methods. On the EDPS dataset, PastNet achieves the highest MS-SSIM score,

significantly better than SimVP. Overall, these results demonstrate that PastNet is highly effective in accurately predicting physical quantities and preserving structural information in various physical datasets, outperforming other state-of-the-art methods such as DLP, PhyDnet, NLDM, and SimVP.

We present the qualitative prediction results of PastNet for various datasets, highlighting its capability to accurately predict future images. Our findings demonstrate that PastNet reliably predicts traffic flow changes in the TrafficBJ dataset. Furthermore, PastNet performs well in the SEVIR weather dataset, as shown in Figure 4, where it accurately predicts high-resolution satellite images and infers future weather changes.

## 4.3 Ablation Study

In this section, we demonstrate PastNet's competitive performance on a wide range of datasets through a series of ablation studies. Table 4 and Figure 7 present the quantitative and qualitative results of these studies for different model structures, respectively. Specifically, **PastNet w/o FPG** removes the FPG module from the PastNet model. **PastNet w/o FPG + UNet** removes the FPG module from the PastNet model and uses UNet as an alternative module. **PastNet w/o FPG + ViT** removes the FPG module from the PastNet model and uses ViT as an alternative module. **PastNet w/o FPG + SwinT** removes the FPG module from the PastNet model and uses SwinT as

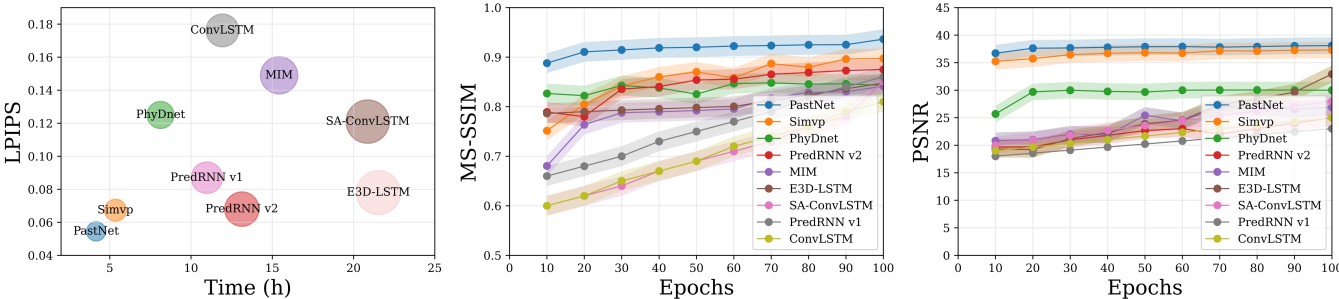

**Figure 5: PastNet outperforms other models in terms of efficiency and convergence rate on the MovingMNIST dataset. Specifically, it achieves the lowest LPIPS score in the shortest training time, as shown on the *Left* side of the figure. In addition, it achieves the highest MS-SSIM and PSNR scores within the same epochs, as depicted in the Middle and *Right* sides of the figure, respectively.**

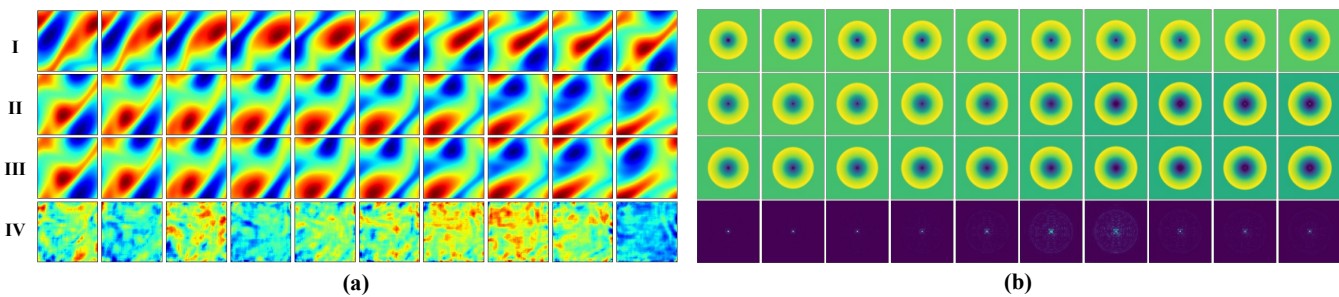

**Figure 6: (a) shows qualitative visualization results for NSE, and (b) displays results for SWE. The input PDE equations are denoted by I, future PDE equations by II, predicted PDE equations by III, and the error between predicted and true results by IV, measured using the relative L2 error metric. The relative L2 errors are 0.0072 for NSE and 0.0007 for SWE.**

an alternative module. **PastNet w/o DST** removes the DST module from the PastNet model. **PastNet** is the base PastNet model, which includes both the DST and FPG modules. All experiments train for 100 epochs.

- PastNet outperforms other models with the lowest MSE, MAE, and highest SSIM, indicating its superior performance in video prediction.
- The inclusion of the DST module in PastNet improves the model's training speed, making it a more efficient option.
- The FPG module is crucial for improving PastNet's performance on the Natural Phenomenon dataset, with Unet as a potential alternative. This demonstrates PastNet's versatility and ability to adapt to different datasets and tasks.

### 4.4 Efficiency and Convergence Rate Analysis

Figure 5 on the left side clearly demonstrates the advantages of PastNet in terms of both time and LPIPS metrics. Notably, PastNet's training time for 100 epochs is only 4.16 hours, considerably faster than other models. Additionally, it achieves outstanding results in terms of LPIPS scores, indicating that PastNet can complete training more efficiently in a shorter amount of time while generating higher-quality images. The middle and right sides of Figure 5 illustrate the rapid improvement of PastNet in SSIM and PSNR metrics. Within approximately 60 epochs, PastNet achieves an SSIM metric of around 0.92, while other models remain below 0.85 at the same

point. After 100 epochs, PastNet reaches a PSNR metric of approximately 38, which is far superior to other models. These results highlight PastNet's superior convergence rate during training, enabling it to rapidly produce high-quality image results. In summary, PastNet is an accurate and efficient model for video prediction, with fast convergence and high-quality results. It represents a promising direction for future research and practical applications.

As shown in Table 5, we select the latest baseline models and compare their efficiency on **TrafficBJ** and **NS equations**. PastNet has significantly lower FLOPs, parameter count, and inference time than SimVP and TAU, while also achieving better MSE and SSIM. In the first experiment, PastNet's FLOPs are 816.32M, parameters are 1.20M, inference time is 0.92s, MSE is 42.93, and SSIM is 0.9876. In the second experiment, FLOPs are 3269.70M, parameters are 1.20M, inference time is 1.02s, MSE is 30.21, and SSIM is 0.9655. Overall, PastNet excels in computational efficiency and prediction accuracy.

### 4.5 Potential for Solving PDE Equations

To investigate the potential of PastNet for solving physical problems, we consider the Navier-Stokes equations, representing viscous incompressible fluids as vorticity on the unit torus, and the Shallow-Water equations, derived from the general Navier-Stokes equations. We focus on their 2D forms.

We use PastNet to solve the Navier-Stokes equations with viscosity $\nu = 10^{-3}$ and a resolution of $64 \times 64$ for training and testing. We

**Table 3: Quantitative prediction results of PastNet compared to baselines on various natural phenomenon datasets. The best result is in bold, and the second-best result is underlined.**

| Dataset | Model | MSE | MAE | MS-SSIM | PSNR |
|---------|-------|-----|-----|---------|------|
| SEVIR | DLP | 300.42 | 140.82 | 0.6772 | 36.59 |
|  | PhyDnet | 97.70 | 72.22 | 0.7137 | 43.03 |
|  | NLDM | 295.93 | 170.73 | 0.6982 | 36.71 |
|  | SimVP | 68.68 | 47.71 | 0.7231 | 49.09 |
|  | PastNet | **66.13** | **44.84** | **0.7568** | **49.78** |
| RDS | DLP | 1.38 | 2.08 | 0.9763 | 44.52 |
|  | PhyDnet | 0.51 | 1.25 | 0.9874 | 47.01 |
|  | NLDM | 1.03 | 1.78 | 0.9594 | 46.56 |
|  | SimVP | 0.15 | 0.67 | 0.9896 | 51.06 |
|  | PastNet | **0.13** | **0.63** | **0.9997** | **51.79** |
| EDPS | DLP | 3.53 | 281.82 | 0.9337 | 42.53 |
|  | PhyDnet | 1.08 | 167.23 | 0.9983 | 45.92 |
|  | NLDM | 2.51 | 237.43 | 0.9455 | 42.83 |
|  | SimVP | **0.93** | **150.11** | 0.9882 | **46.25** |
|  | PastNet | 0.94 | 168.53 | **0.9991** | 45.98 |
| FS | DLP | 7.78 | 426.12 | 0.9266 | 38.38 |
|  | PhyDnet | 4.41 | 327.32 | 0.9423 | 40.47 |
|  | NLDM | 7.21 | 411.14 | 0.9392 | 38.62 |
|  | SimVP | 3.01 | 261.19 | 0.9647 | 42.03 |
|  | PastNet | **2.19** | **222.08** | **0.9861** | **43.24** |

**Table 4: Ablation study results with different model structures on the SEVIR dataset.**

| Model | MSE | MAE | SSIM | Time (h) |
|-------|-----|-----|------|----------|
| PastNet w/o FPG | 133.8 | 103.6 | 0.67 | 3.19 |
| PastNet w/o FPG + UNet | 115.4 | 99.35 | 0.61 | 16.54 |
| PastNet w/o FPG + ViT | 287.3 | 139.5 | 0.49 | 23.87 |
| PastNet w/o FPG + SwinT | 321.8 | 158.7 | 0.54 | 24.12 |
| PastNet w/o DST | 192.3 | 118.5 | 0.64 | 4.92 |
| PastNet | **73.21** | **52.31** | **0.74** | **4.16** |

**Table 5: Performance comparison of different models.**

| | FLOPs (M) | Param (M) | Inference Time (s) | MSE x 100 | SSIM |
|---|-----------|-----------|---------------------|-----------|------|
| PastNet | **816.32** | **1.20** | **0.92** | **42.93** | **0.9876** |
| SimVP | 980.52 | 13.83 | 0.97 | 43.32 | 0.9822 |
| TAU | 3918.79 | 55.30 | 1.24 | 43.01 | 0.9854 |
| PastNet | **3269.70** | **1.20** | **1.02** | **30.21** | **0.9655** |
| SimVP | 5217.71 | 14.43 | 1.22 | 35.94 | 0.9342 |
| TAU | 20844.64 | 57.69 | 1.33 | 32.34 | 0.9453 |

use the flow field at 10 input time steps to predict the flow field at 10 future time steps ($10_{timesteps} \mapsto 10_{timesteps}$). For the Shallow-Water equations, we set the resolution to $128 \times 128$ for training and testing and use the flow field at 50 input time steps to predict the flow field at 50 future time steps ($50_{timesteps} \mapsto 50_{timesteps}$). We train for 200 epochs and record the metrics MSE, MAE, and time per epoch. The quantitative and qualitative results are in Table 6 and Figure 6, respectively.

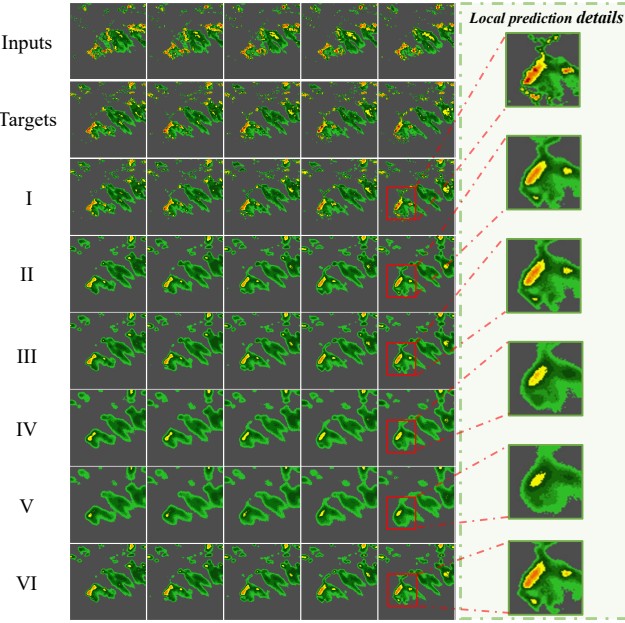

**Figure 7: *Inputs*: The input weather sequences; *Targets*: The future real weather sequences; *I*: PastNet predicts the weather sequences; *II*: PastNet w/o FPG predicts the weather sequences; *III*: PastNet w/o FPG + UNet predicts the weather sequences; *IV*: PastNet w/o FPG + ViT predicts the weather sequences; *V*: PastNet w/o FPG + SwinT predicts the weather sequences; *VI*: PastNet w/o DST predicts the weather sequences.**

**Table 6: Performance metrics for solving PDE equations.**

| PDE | Evaluation Criterion | | |
|-----|------|-----|--------------------|
|  | MSE | MAE | Time per epoch (s) |
| NSE | 0.3021 | 28.4952 | 218 |
| SWE | 0.0185 | 8.0391 | 223 |

The results in Table 6 show that PastNet has potential for solving PDE equations, especially for the SWE equation. It achieves lower MSE and MAE values for the SWE equation than for the NSE equation, while the time per epoch is similar for both equations.

## 5 CONCLUSION

In this paper, we investigate the problem of spatio-temporal video prediction and propose a novel method named PastNet to tackle the problem. The key insight of our PastNet is to incorporate a spectral convolution operator in the Fourier domain, which effectively introduces inductive biases from the underlying physical laws. Moreover, we introduce both local feature discretization and intrinsic dimensionality estimation to reduce the computational costs with accuracy retained. Extensive experiments on a range of popular datasets show that the proposed PastNet is more effective and efficient than state-of-the-art techniques. In future works, we would develop more effective video prediction techniques by introducing high-level physical domain knowledge in various fields.

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
