# OpenReview forum: "PastNet: Introducing Physical Inductive Biases for Spatio-temporal Video Prediction"
_acmmm.org/ACMMM/2024/Conference — MM2024 Poster_

### Official Review · Reviewer_cQR6 · 2024-05-08

**Rating:** 3
**Confidence:** 3

**Summary:**

This paper investigate spatio-temporal video prediction task, generating future video frames based on existing observation streams. Authors focus on physical knowledge embedded within videos and computational costs, proposing the PastNet. Extensive experiments on  spatio-temporal video benchmarks demonstrate the effectiveness.

**Strengths:**

+ Spatio-temporal video prediction is In line with the theme of the ACM MM.
+ Paper is well-organized and the layout is orderly.
+ Authors initiate a new insight on spatio-temporal video prediction with physical inductive biases.
+ The experiments are abundant and the metrics are comprehensive

**Limitations:**

If authors can solve all issues I concerned, $\textbf{I am willing to raise my score according to rebuttal.}$
- Authors declare that their method has a better performance and show their prediction results on the SEVIR dataset in Figure 5. However, the comparison with prediction results of other SOTA is absent. It's not convincing. Authors are suggested to reveal their results like Figure 6. The same applies to Figure 3, 4 and Figure 1, 2 (In appendix).
- Computational costs are emphasized by authors. I can see authors show training in Table 4 while complexity is not discussed. What about the parameter number and Flops of the PastNet?
- According to the results in Table 1, 2. The improvement of PastNet is not clear when compared with SimVP.
- Authors should compare their method with recent method such as:
    - Ning et al. 2023. MIMO is all you need: A strong multi-in-multi-out baseline for video prediction. In AAAI.
    - Tan et al. 2023. Openstl: A comprehensive benchmark of spatio-temporal predictive learning. In NeurIPS.
- Some long and difficult sentences make it difficult to understand and writing mistakes including:
    - expression of disunity “spatiotemporal prediction” or “spatio-temporal prediction” in the main paper.
    - Subsection ?? at the end of the appendix.

**Suitability:**

2

---

### Official Review · Reviewer_LhCq · 2024-05-09

**Rating:** 4
**Confidence:** 3

**Summary:**

This paper studies an important problem of spatio-temporal data prediction. The authors propose a physics-assisted spatio-temporal network called PastNet to perform high-quality spatio-temporal data synthesis. The proposed PastNet considers the inherent physical knowledge embedded features and reduces the computational overhead. The extensive experiments show the superiority of the proposed method.

**Strengths:**

1. The paper is well-written and easy to follow. The organization of the paper is clear.

2. The proposed PastNet incorporates a spectral convolution operator in the Fourier domain to learn the inherent physical knowledge, which provides explainability.

3. The authors design a memory bank with the estimated intrinsic dimensionality to reduce the computational cost.

4. Extensive experiments offer the evidence of the effectiveness of the proposed method.

**Limitations:**

1. It is suggested to illustrate the relationship between the proposed method (or the studied problem) and multimedia, which seems to be a little bit unclear.

2. It would be better to provide a discussion to show how the proposed FPG module is explainable. In addition, how does the proposed FPG module introduce physical inductive biases for the studied problem?

3. Compared to Simvp, the improvement of PastNet is limited in terms of computational overhead and accuracy, especially on EDPS. It is suggested to provide the advantages of PastNet. In addition, it is suggested to discuss the computational complexities (space and time complexities) of PastNet and baselines, to demonstrate the efficiency of PastNet theoretically.

4. The evaluation metrics are not specified for spatio-temporal data prediction, which may neglect the specific and important spatio-temporal characteristics. More evaluation metrics are suggested to be included. Please refer to the following paper.

    [1] Sai et al., TIME WEAVER: A Conditional Time Series Generation Model, ICML 2024.

6. It is suggested to investigate more related work as follows.

    [2] Lin et al., Self-Attention ConvLSTM for Spatiotemporal Prediction, AAAI 2020.

    [3] Liang et al., GeoMAN: Multi-level Attention Networks for Geo-sensory Time Series Prediction, IJCAI 2018.

    [3] Wang et al., Multi-task adversarial spatial-temporal networks for crowd flow prediction, CIKM 2020.

    [4] Zhang et al., Predicting citywide crowd flows using deep spatio-temporal residual networks, Artificial Intelligence 2018.

    [5] Xu et al., PIGAT: Physics-Informed Graph Attention Transformer for Air Traffic State Prediction, TITS 2024.

7. The studied problem seems similar to spatio-temporal data prediction, especially physics-informed spatio-temporal data prediction. It would be better to show the connection and difference between the studied problem and spatio-temporal data prediction. There are some physics-informed spatio-temporal data prediction methods [5, 6], especially in terms of traffic prediction, which are suggested to be compared for fairness.

    [6] Ji et al., STDEN: Towards physics-guided neural networks for traffic flow prediction, AAAI 2022.

**Suitability:**

2

---

### Official Review · Reviewer_9aVr · 2024-05-24

**Rating:** 3
**Confidence:** 2

**Summary:**

The proposed PastNet model introduces both Fourier-based Physics-Guided (FPG) and Discrete Spatio-Temporal (DST) modules, which effectively improve spatio-temporal video prediction by incorporating physical principles and intrinsic dimensionality estimation.

**Strengths:**

* The paper is well-structured and well-written.
* The method and experiments are persuasive.

**Limitations:**

**Weakness**
1. The claimed efficiency is limited if it is only reflected in training time. SimVP seems simpler and more efficient.
2. The Fourier-based token mixer is not novel.
3. The comparison experiments are unfair.

**Questions**
1. Compared to other algorithms, what are the parameter count and FLOPs of PastNet? What is its inference speed on real devices?
2. The experiments do not compare against the latest algorithms. It seems the most advanced work cited, SimAP, is from 2022. However, to my knowledge, there have been some recent works in 2023 and very recently [1].
3. The experimental dataset settings are inconsistent with those of the compared algorithms. Why?
4. In Table 2, citations and publication years for the compared algorithms can be added to help readers locate them.

[1] Tan, Cheng, et al. "Openstl: A comprehensive benchmark of spatio-temporal predictive learning." Advances in Neural Information Processing Systems 36 (2023): 69819-69831.

**Suitability:**

2

---

### Official Review · Reviewer_gE4e · 2024-05-25

**Rating:** 4
**Confidence:** 3

**Summary:**

The paper investigates the challenges of spatio-temporal video prediction, which involves generating future videos based on historical data streams. The paper proposes the novel approach of  inherent physical knowledge embedded within videos, and  introduce a novel approach called Physics-assisted Spatio-temporal Network (PastNet) for generating high-quality video prediction.

**Strengths:**

The paper opens a new perspective to link spatio-temporal video prediction with physical inductive bias so as to augment the model with the data itself.The proposed PastNet not only uses convolution operators in the spectral space to incorporate the physical prior, but also discretises the local features using memories with estimated intrinsic dimensions to improve the efficiency of high-resolution video prediction.The paper declares that PastNet has high performance and efficiency.

**Limitations:**

1,the paper used for comparison should be more recent, e.g., Modular Action Concept Grounding in Semantic Video Prediction, 2022. comparing Correspondences: Video Prediction with Correspondence-wise Losses, 2022. MMVP: Motion-Matrix-Based Video Prediction 2023.
2,In the DST module mentioned in the paper, discrete libraries are built to save computational costs. Generally libraries are created to make better use of historical information, the article is how to achieve cost reduction by creating discrete libraries.

**Suitability:**

3

---

### Meta-Review · Area_Chair_hDef · 2024-07-04

**Recommendation:** Accept (Poster)
**Confidence:** 4

**Metareview:**

The key novelty of the paper is introducing physical inductive biases for spatiotemporal video prediction. This, coupled with intrinsic dimensionality estimation with a memory bank, reduces computational time. The paper received 3 BAs and 1 BR after rebuttal. The remaining concerns are persuasion of visualization (Reviewer cQR6), more evaluation metrics (Reviewer LhCq), and addressing the connection and difference between the studied problem and spatio-temporal data prediction (Reviewer LhCq). The reviewers all liked the paper's new insight. The AC likes the paper's contribution and thinks the remaining concerns are addressable in a minor revision and, hence, recommends acceptance.